# KAC: Kolmogorov-Arnold Classifier for Continual Learning

## Abstract

Continual learning requires models to train continuously across consecutive tasks without forgetting. Most existing methods utilize linear classifiers, which struggle to maintain a stable classification space while learning new tasks. Inspired by the success of Kolmogorov-Arnold Networks (KAN) in preserving learning stability during simple continual regression tasks, we set out to explore their potential in more complex continual learning scenarios. In this paper, we introduce the Kolmogorov-Arnold Classifier (KAC), a novel classifier developed for continual learning based on the KAN structure. We delve into the impact of KAN's spline functions and introduce Radial Basis Functions (RBF) for improved compatibility with continual learning. We replace linear classifiers with KAC in several recent approaches and conduct experiments across various continual learning benchmarks, all of which demonstrate performance improvements, highlighting the effectiveness and robustness of KAC in continual learning.

## 1 Introduction

Deep learning models are typically trained on a fixed dataset in a single session, achieving impressive performance on various static tasks. In contrast, real-world scenarios continuously evolve, necessitating models that can learn incrementally from a data stream. However, in such scenarios, these models often encounter a significant challenge known as catastrophic forgetting (French, 1999). Continual learning (De Lange et al., 2022; Belouadah et al., 2021; Parisi et al., 2019; Golab & Özsu, 2003) investigates how to effectively train models in such dynamic environments with sequential data exposure, aiming to adapt and avoid forgetting over time.

Class incremental learning (CIL) (Rebuffi et al., 2017), as a key challenge in continual learning, has garnered extensive research interest. It involves the continuous introduction of new classes with ongoing tasks, requiring the model to conduct classification on all encountered classes after training on new tasks. Most CIL methods retain exemplars and employ techniques, such as knowledge distillation (Rebuffi et al., 2017; Douillard et al., 2020; Wen et al., 2024) or dynamic architectures (Chen & Chang, 2023; Douillard et al., 2022; Yan et al., 2021; Kim et al., 2024b), to mitigate forgetting. With the rise of pre-trained models, numerous studies (McDonnell et al., 2024; Zhang et al., 2023) have explored their applications in CIL, achieving impressive results. Among these, prompt-based approaches (Wang et al., 2022d;c; Smith et al., 2023; Gao et al., 2024b) have attracted considerable attention.

Among existing methods, McDonnell et al. (2024); Goswami et al. (2024); Yu et al. (2020) focused on feature space design through carefully crafted classifiers and training or inference strategies, achieving excellent performance. These studies demonstrate that a well-structured feature space can effectively mitigate forgetting because a stable distribution is crucial for continual classification tasks while the design of classifiers is essential for constructing the feature space and reducing forgetting in continuous tasks. However, most existing approaches (Gao et al., 2024b; Zhou et al., 2024; Smith et al., 2023) utilize linear classifiers or nearest class mean classifiers (NCM) (Rebuffi et al., 2017), with limited research focused on developing a specific classifier for CIL to effectively mitigate catastrophic forgetting. Therefore, designing an efficient classifier that replaces the existing simple classifiers and enhances current approaches will significantly advance the development of CIL.

Recently, a novel model architecture, Kolmogorov–Arnold Networks (KAN) (Liu et al., 2024), has been proposed, demonstrating natural effectiveness in continual learning. The authors compare

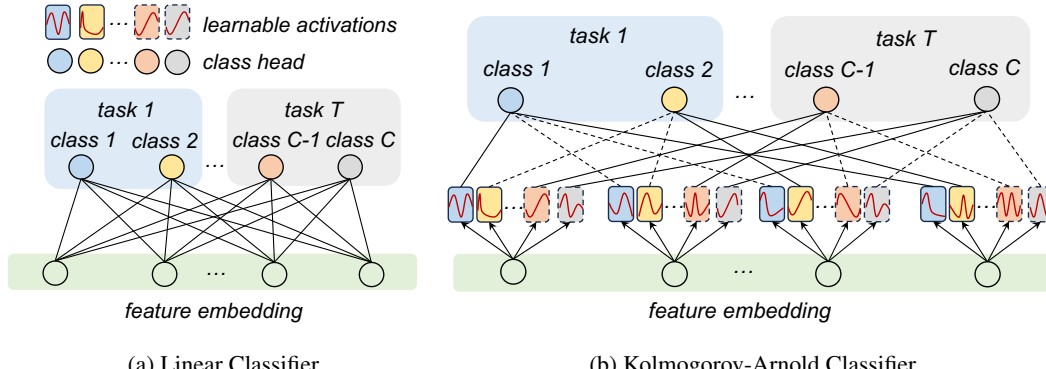

(a) Linear Classifier  (b) Kolmogorov-Arnold Classifier

Figure 1: Brief comparison between conventional linear classifier and our Kolmogorov-Arnold classifier. The solid lines represent activated weights, while the dashed ones represent suppressed weights. Conventional linear classifiers activate each weight equally across all tasks, whereas our Kolmogorov-Arnold Classifier learns class-specific learnable activations for each channel across all categories, minimizing forgetting caused by irrelevant weight changes.

KAN with Multi-Layer Perceptrons (MLP) (Hornik et al., 1989) on a toy continual 1D regression problem, which requires the model to fit 5 Gaussian peaks sequentially. KAN exhibits superior performance, effectively mitigating catastrophic forgetting, attributed to the locality of splines and inherent local plasticity. This locality allows KAN to identify relevant regions for re-organization while maintaining stability in other areas during sequential tasks (Liu et al., 2024). These findings motivate us to explore the applications of KAN in more challenging CIL tasks.

In this paper, we present the Kolmogorov-Arnold Classifier (KAC), a plug-and-play classifier for Continual Learning based on the KAN architecture. Leveraging the Kolmogorov-Arnold representation theorem (Kolmogorov, 1961), we incorporate learnable activation functions on the edges of the classifiers. We find that the conventional KAN with B-spline functions struggles with high-dimensional data, leading to inadequate model plasticity, which may weaken the models' plasticity when directly introduced as a classifier. This limitation forces models to undergo excessive updates when learning new tasks, resulting in significant forgetting.

To address this, we explore spline functions and identify Radial Basis Functions (RBF) as an effective alternative for continual learning. By utilizing RBF in our KAC, we enhance the model's ability to adapt CIL while minimizing forgetting. Thanks to these learnable spline activations, the KAC allows the model to select specific activation ranges of interest for each channel while preserving the distribution of other parts, and RBF makes it more compatible with CIL. As shown in Fig. 1b, these learnable activations help the model select interesting parts of each channel and activate them for determination rather than activating all edges like a simple linear classifier in Fig. 1a. This brings notable benefits to class incremental learning. When new tasks arrive, the learnable activation functions assist the model in selecting relevant parts of each channel for updating. This prevents the drift of irrelevant features during the training process for the new tasks. Meanwhile, the deactivated portions of the old tasks remain unaffected by these updates, reducing the forgetting of old tasks.

To demonstrate the superiority of KAC, we conduct experiments on several prompt-based continual learning approaches, which are built upon a pre-trained backbone where the classifiers play a key role in these approaches. The models employing our method achieve significant improvement across various CIL scenarios on multiple datasets by simply replacing the linear classifier with our KAC without making any other modifications or hyperparameter adjustments. Additionally, experiments conducted in the Domain Incremental Learning (DIL) (Wang et al., 2022b) setting reveal that our method can also improve performance, demonstrating its effectiveness and robustness.

Our main contributions can be summarised as follows:

- We explore the application of Kolmogorov-Arnold Networks (KAN) in continual learning and analyze its weaknesses when employed in continual learning and how to enhance its compatibility with such tasks.

- We introduce the Kolmogorov-Arnold Classifier (KAC), a novel continual classifier based on the KAN structure with Radial Basis Functions (RBF) as its basis functions. KAC enhances the stability and plasticity of CIL approaches.

- We integrate our KAC into various approaches and validate their performance across multiple continual learning benchmarks. The results demonstrate that KAC can effectively reduce forgetting in these methods.

## 2 RELATED WORK

**Class Incremental Learning** aims to learn a sequence of classification tasks sequentially, where the number of classes increases with each task. The primary challenge in it is catastrophic forgetting(McCloskey & Cohen, 1989). Several studies work on it and they can be broadly categorized into three main strategies: regularization-based, structure-based, and replay-based methods. Regularization-based methods reduce forgetting by employing knowledge distillation techniques(Wen et al., 2024; Yang et al., 2022; Douillard et al., 2020) or imposing constraints on key model parameters(Kang et al., 2022; Kirkpatrick et al., 2017). Structure-based methods(Chen & Chang, 2023; Wang et al., 2022a; Douillard et al., 2022) mitigate forgetting through dynamic network architectures. Replay-based methods retain a small portion of old data(Jeeveswaran et al., 2023; Rebuffi et al., 2017) or use auxiliary models(Kim et al., 2024a; Gao & Liu, 2023; Shin et al., 2017) to generate synthetic data, which are combined with new-class data to update the model.

**CIL with Pre-trained Models** have demonstrated their competitive performance in Class Incremental Learning due to their strong transferability. Techniques such as LAE (Gao et al., 2023) and SLCA (Zhang et al., 2023) enhance model adaptation through EMA-based updates and dynamic classifier adjustments. RanPAC (McDonnell et al., 2024) employs random projection to improve continual learning, while EASE (Zhou et al., 2024) focuses on optimizing task-specific, expandable adapters to enhance knowledge retention.

Benefiting from parameter-efficient tuning in NLP, prompt-based methods have achieved promising results in Class Incremental Learning. These approaches utilize adaptive prompts to guide frozen transformer models, facilitating efficient task-specific learning without modifying encoder parameters. Techniques like L2P (Wang et al., 2022d), DualPrompt (Wang et al., 2022c), S-Prompts (Wang et al., 2022b), CODA-Prompt (Smith et al., 2023), HiDe-Prompt (Wang et al., 2024), and CPrompt (Gao et al., 2024a) introduce diverse prompt selection strategies to improve task learning, knowledge retention, and model robustness.

**Kolmogorov-Arnold Networks** (KAN) (Li, 2024) is a novel network architecture based on the Kolmogorov-Arnold representation theorem (Kolmogorov, 1961). It represents multivariate functions as combinations of multiple univariate functions and uses nonlinear spline functions for approximation. Some explorations focus on how to apply KAN to solve scientific problems (Koenig et al., 2024; Bozorgasl & Chen, 2024; Howard et al., 2024), while others seek various basis functions to enhance performance (Aghaei, 2024; Bozorgasl & Chen, 2024; Li, 2024). Many works (Bresson et al., 2024; De Carlo et al., 2024; Genet & Inzirillo, 2024; Malashin & Mikhalkova, 2024) apply KAN across various fields and investigate how to effectively leverage its advantages in these domains. These studies encourage us to explore the application of KAN in continual learning.

## 3 METHOD

### 3.1 PRELIMINARIES

**Class Incremental Learning.** In Class Incremental Learning (CIL), a model needs to learn classes step by step. At each step $t$, the model needs to learn the classes specific to that step, denoted as $\mathcal{Y}_t$, with only access to the current dataset $D_t = \{(x_t^i, y_t^i)\}_{i=1}^{n_t}$, where $x_t^i$ represents an input image and $y_t^i$ is its corresponding label. A key challenge in CIL is how to maintain the stability of the model to avoid catastrophic forgetting (French, 1999) while learning new tasks. With a model consisting of a backbone $F$, and a classifier $h \in \mathbb{R}^{n \times C}$, where $n$ denotes the embedding dimension and $C$ represents the total number of learned classes, the model is tasked with predicting the class label $y = h(F(x)) \in \mathcal{Y}$ for test samples from new classes as well as samples from previously encountered tasks.

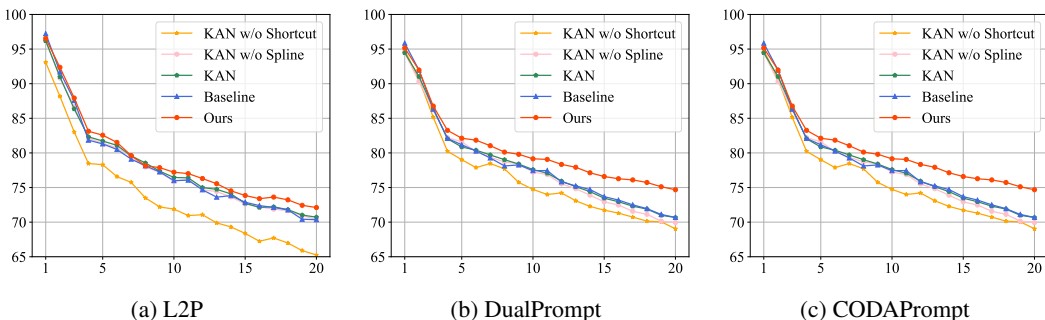

(a) L2P                   (b) DualPrompt                   (c) CODAPrompt

Figure 2: Comparison of the accuracy curves of three recent approaches with different classifiers in the ImageNet-R 20-step scenario. The x-axis represents the increasing number of tasks, while the y-axis shows the corresponding test accuracy at each step. The Baseline indicates performance with a conventional linear classifier, while the other curves represent results with ablated KAN classifiers and our Kolmogorov-Arnold Classifier.

**Kolmogorov–Arnold Networks.** Kolmogorov–Arnold Networks (KAN) (Liu et al., 2024) is a novel model architecture that serves as a promising alternative to multi-layer perceptrons (MLPs) (Haykin, 1998; Hornik et al., 1989). While MLPs rely on the Universal Approximation Theorem (UAT) (Hornik et al., 1989), KANs are inspired by the Kolmogorov-Arnold representation Theorem (KAT) (Kolmogorov, 1961). KAT posits that any multivariate continuous function $f(x)$ defined on a bounded domain can be expressed as a finite composition of univariate continuous functions through addition. The Kolmogorov-Arnold representation theorem can be written as:

$$f(x) = f(x_1, x_2, ..., x_n) = \sum_{q=1}^{2n+1} \Phi_q \Big( \sum_{p=1}^{n} \phi_{q,p}(x_p) \Big), \tag{1}$$

in which $\Phi_q$ and $\phi_{q,p}$ are univariate functions for each variable. KAN parametrizes the $\phi_{q,p}$ and $\Phi_q$ as B-spline curves, with learnable coefficients of local B-spline basis functions $B(x)$ (Qin, 1998). In practice, a residual connection, consisting of a linear function with activation $b(x) = silu(x) = x/(1 + e^{-x})$, is linearly combined with the B-spline curve $spline(x) = \sum_i \omega_i B_i(x)$ to form the final $\phi$:

$$\phi(x) = \omega_b b(x) + \omega_s spline(x), \tag{2}$$

where the $\omega_b$ and $\omega_s$ represent the linear functions that control the overall magnitude of the activation function. Consequently, a KAN layer can be expressed as:

$$x_{l+1} = \underbrace{\begin{pmatrix} \phi_{l,1,1}(.) & \phi_{l,1,2}(.) & \cdots & \phi_{l,1,n_l}(.) \\ \phi_{l,2,1}(.) & \phi_{l,2,2}(.) & \cdots & \phi_{l,2,n_l}(.) \\ \vdots & \vdots & \ddots & \vdots \\ \phi_{l,n_{l+1},1}(.) & \phi_{l,n_{l+1},2}(.) & \cdots & \phi_{l,n_{l+1},n_l}(.) \end{pmatrix}}_{\Phi_l} x_l. \tag{3}$$

The $x_l$ and $x_{l+1}$ represent the input and output of a KAN layer, while $\phi_l$ is the 1D univariate function matrix for each layer. The KAN networks are constructed by stacking multiple KAN layers.

### 3.2 CONVENTIONAL KAN LAYER IS NOT A GOOD CONTINUAL CLASSIFIER

In Liu et al. (2024), the authors present experimental results from a toy 1D regression task, demonstrating that the locality of splines can inherently avoid catastrophic forgetting. This insight inspires us to introduce KAN to CIL. A straightforward way to leverage the locality of KAN is directly utilizing a KAN layer to develop a continual classifier, replacing the linear classifier in CIL methods. To

achieve this, we simply replace the linear classifier $h(x)$ with a KAN layer that has an input dimension of $d$ and an output dimension of $C$. We compared their performances across several baseline methods. The experimental results are shown in Fig. 2, demonstrating that the simple substitution of replacing the linear classifier with a KAN layer does not lead to any improvement, even achieving worse performance.

We decompose the KAN layer into two parts: the residual connection $b(x)$ and the B-spline curve $spline(x)$ and individually replace the linear classifier with these two components to investigate why directly introducing the KAN layer increases forgetting. A surprising finding is that the B-spline functions lead to a severe performance drop across all baselines.

To understand why the B-spline curve replacing the conventional linear classifier leads to severe forgetting, we need to delve deeper into the differences between linear layers and splines. In high-dimensional complex data, spline functions encounter the curse of dimensionality (COD) (Hammer, 1962); as the data dimensionality increases, the model struggles with data approximation (Köppen, 2002; Montanelli & Yang, 2020; He, 2023). This is because splines cannot effectively model the compositional structure present in the data, while linear classifiers benefit from their fully connected structure, allowing them to learn this structure effectively (He & Xu, 2023). Although KAN networks mitigate COD through approximation theory (Liu et al., 2024) by stacking KAN layers, approximating high-dimensional function remains a challenging problem for a single spline layer, whereas it is relatively straightforward for conventional linear classifiers.

It is precisely the weak fitting ability of B-spline functions on high-dimensional data that leads to severe forgetting when it is introduced into CIL. In CIL, a network typically consists of a backbone $F$ that encodes images to feature embeddings and a classification head $h$, which serves as a high-dimensional projection mapping the embeddings to class probabilities. Most methods accommodate new classes by adding classifiers while sharing the backbone across all tasks. The final logits $l$ for classification are always calculated as:

$$l = h(F(x)), h = [h_1, h_2, \cdots, h_t]. \tag{4}$$

To prevent significant forgetting caused by changes in the backbone that affect the feature space, the model must maintain stable backbone parameters during training on new tasks. Consequently, many methods use regularization techniques to restrict changes in feature embeddings (Li & Hoiem, 2017; Kim et al., 2024b; Wen et al., 2024; Yang et al., 2022). However, due to the limited approximation capability of a single B-spline layer, the model requires more extensive updates to the backbone parameters compared to conventional linear classifiers to achieve good performance on new tasks. This extensive updating can severely disrupt the feature space, leading to pronounced forgetting.

Based on the above analysis, we believe that the weak fitting ability of a single B-spline function prevents the model from leveraging the locality of the KAN layer. Therefore, we need to enhance the spline function's fitting ability to adapt the KAN structure to CIL tasks. Lin et al. (2017); Lai & Shen (2021) indicates that, in specific senses, a shallow KAT-based layer can break the COD problem when approximating high-dimensional functions through designed basis functions with particular compositional structures, motivates us to explore the types of basis functions that are compatible with CIL.

### 3.3 Radial basis function is great for class incremental learning

Several studies (McDonnell et al., 2024; Zhuang et al., 2023; Yu et al., 2020) assume that the classification space follows a Gaussian space and develop approaches based on this premise, achieving excellent performance. It suggests that building a Gaussian classification space can help models effectively learn new tasks while combating catastrophic forgetting. Can we find a kind of basis function in this sense that allows a KAT-based layer function as a continual classifier, addressing the COD problem and benefiting CIL? The answer is yes!

FastKAN (Li, 2024) proves that the B-splines basis function in KAN (Liu et al., 2024) can be well replaced by Radial Basis Functions (RBF) (Buhmann, 2000; Orr et al., 1996). We find this substitution brings more benefits to CIL when KAN is introduced as a continual classifier as shown

later. A KAN layer with RBF is represented as:

$$f(x) = \sum_{p=1}^{n} \Phi_p \sum_{i=1}^{N} \omega_{p,i} \phi(||x_p - c_i||),$$ (5)

where $c_i$ represents a series of center points evenly distributed within a specific range, with $N$ denoting the total number of $c_i$. And $\phi(x)$ is an RBF served as the basis functions whose value solely depends on the distance between input $x$ and center point $c_i$. The term $\omega_{p,i}$ denotes the weight for each $\phi$. A Gaussian function with covariance $\sigma_i$ can be chosen as $\phi$ while it's defined as:

$$\phi(||x_p - c_i||) = \exp\left( - \frac{(x_p - c_i)^2}{2\sigma_i^2} \right).$$ (6)

While introducing the Gaussian RBF function as the basis function of KAN demonstrates faster evaluation speeds and enhanced performance, as shown in Li (2024), an inherent Gaussian structure is also established with it, which can serve as an effective compositional structure for CIL scenarios.

With a series of Gaussian distributions $\mathcal{N}$ centered at $c = [c_1, c_2, \cdots, c_N]$ assumed to be independent, the activation function for each dimension is formed by combining $N$ independent Gaussian distributions, and the distribution of each dimension after activation can be represented as:

$$\sum_{i=1}^{N} \omega_{p,i} \phi(||x_p - c_i||) \sim \omega_{p,1} \mathcal{N}(c_1, \sigma_1) + \omega_{p,2} \mathcal{N}(c_2, \sigma_2) + \cdots + \omega_{p,N} \mathcal{N}(c_N, \sigma_N)$$

$$= \mathcal{N}\left( \omega_{p,1} c_1 + \cdots + \omega_{p,N} c_N, \omega_{p,1}^2 \sigma_1^2 + \cdots + \omega_{p,N}^2 \sigma_N^2 \right).$$ (7)

The second equation is based on the additivity of the Gaussian distribution (Lemons & Langevin, 2002). We can easily derive that, thanks to the introduction of Gaussian RBF functions, the features of $p$th dimension in the KAN layer, after the activation function, follow a Gaussian distribution with mean $\mu_p = \sum_{i=1}^{N} \omega_{p,i} c_i$ and variance $\sigma_p = \sum_{i=1}^{N} \omega_{p,i}^2 \sigma_i^2$. This results in the final prediction for each class being represented as the sum of a set of Gaussian distributions, represented as:

$$f(x) = \sum_{p=1}^{n} \Phi_p \cdot \exp\left( - \frac{(x_p - \mu_p)^2}{2\sigma_p^2} \right).$$ (8)

When we simply define $\Phi_p$ as a learnable weight for each dimension, it is evident that the resulting function form conforms to the Gaussian Process (GP) with first-order additive kernels defined in Duvenaud et al. (2011). This structure is consistently easy to fit for classification tasks and possesses a strong long-range structure to effectively address the COD problem when approximating high-dimensional functions (Duvenaud et al., 2011). With functions like this serving as the basis functions for continual classifiers, it not only projects each channel of the feature into a Gaussian space but also allows the model to select an interested range for each channel tailored to different classes.

## 3.4 KOLMOGOROV–ARNOLD CLASSIFIER FOR CIL

The above analysis demonstrates that the KAN layer with RBF can benefit CIL, motivating us to introduce our Kolmogorov-Arnold Classifier (KAC), which can be integrated into any CIL approach by replacing the conventional linear classifier with it.

An overview of the KAC is shown in Fig. 3. The KAC firstly regularizes the feature distribution with a Layer Normalization $\mathcal{LN}$, resulting in a normalized embedding $\mathcal{LN}(F(x)) = [x_1', x_2', \cdots, x_n']$. After that, it incorporates a KAN layer that includes $N$ Gaussian Radial Basis Functions centered at $c = [c_1, c_2, \cdots, c_N]$. With the basis function $\phi$ is like defined in eq. 6, the logit $l$ is then calculated as:

$$l = KAC(F(x)) = \text{diag}\left( W_C \cdot \Phi\left( \mathcal{LN}(F(x)) \right) \cdot W_q \right),$$ (9)

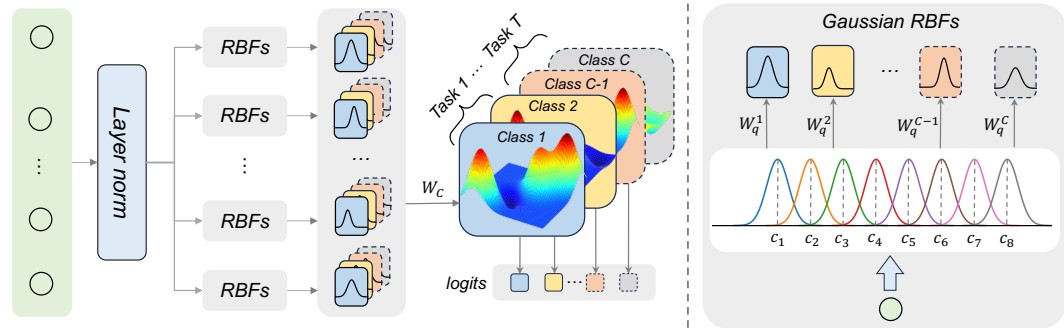

Figure 3: An overview of the pipeline of the proposed Kolmogorov-Arnold Classifier. For the input feature embeddings, we first normalize them using a layer normalization, then pass them through a set of RBFs that activate them to learnable Gaussian distributions. Finally, we weight all channels with $W_C$ to obtain the decision space for each class. The right side shows the process of Gaussian RBFs, which map univariate variables to different Gaussian distributions centered at various points and weight these distributions with $W_q^c$ to derive the final activation distribution for each channel across all classes. The output logits are sampled based on the channel values within the distribution of each class. As tasks increase, new classes can be accommodated by simply expanding $W_C$.

where $\mathrm{diag}(.)$ represents extracting the diagonal elements of a matrix and the $\Phi\Big(\mathcal{LN}\big(F(x)\big)\Big)$ is the learnable Gaussian RBF like:

$$\Phi\Big(\mathcal{LN}\big(F(x)\big)\Big) = \begin{pmatrix} \phi(||x_1' - c_1||) & \phi(||x_1' - c_2||) & \cdots & \phi(||x_1' - c_N||) \\ \phi(||x_2' - c_1||) & \phi(||x_2' - c_2||) & \cdots & \phi(||x_2' - c_N||) \\ \vdots & \vdots & \ddots & \vdots \\ \phi(||x_n' - c_1||) & \phi(||x_n' - c_2||) & \cdots & \phi(||x_n' - c_N||) \end{pmatrix}, \quad (10)$$

in which $n$ is the dimensionality of the input embedding and $W_C \in \mathbb{R}^{C \times n}$ is a learnable weight matrix that serves as an output linear function to predict the probability for each class, corresponding to the $\Phi_p$ in conventional KAN, while the $W_q \in \mathbb{R}^{N \times C}$ corresponds to the $\phi_{p,q}$ in conventional KAN to serve as the univariate learnable activation for each channel for every class. In practice, the $W_C$ and $W_q$ can be consolidated into a single weight matrix $W \in \mathbb{R}^{C \times (N \times n)}$, from which the final logit is directly predicted using the basis functions $\phi$. The KAC is then represented as:

$$KAC\big(F(x)\big) = W \cdot \mathrm{reshape}\bigg(\Phi\Big(\mathcal{LN}\big(F(x)\big)\Big)\bigg). \quad (11)$$

The $\mathrm{reshape}(.)$ function flattens the $N \times n$ matrix into a 1D vector to facilitate calculations with $W$.

In a CIL scenario, $T$ tasks arrive sequentially with class counts $[C_1, C_2, \cdots, C_T]$. KAC expands $W$ to accommodate new classes, similar to conventional classifiers (Smith et al., 2023). At the $t$th step, there is an old classification matrix $W^{t-1} \in \mathbb{R}^{(N \times n) \times C_{old}}$, where $C_{old} = C_1 + C_2 + \cdots + C_{t-1}$, and a new matrix $W^t \in \mathbb{R}^{(N \times n) \times C_t}$, with the final $W$ after the $t$th step being the concatenation of these two matrices.

## 4 EXPERIMENTS

### 4.1 BENCHMARKS & IMPLEMENTATIONS

**Benchmarks.** We evaluate the CIL scenario and further validate the robustness of our method in Domain Incremental Learning (DIL) (Wang et al., 2022b). For CIL, we conduct experiments on two commonly used datasets, ImageNet-R (Hendrycks et al., 2021) and CUB200 (Wah et al., 2011), each containing 200 classes. Starting with 0 base classes, all classes are separated into 5, 10, 20, and 40 steps to feed the model for training sequentially. For DIL, following Sprompt (Wang et al.,

Table 1: Results on ImageNet-R dataset. We report the accuracy of the last task on CIL scenarios of 5, 10, 20, and 40 steps and make comparisons on various approaches, evaluating the results with a linear classifier (baseline) and with our KAC. It demonstrates that our KAC consistently improves their performance, especially in long-sequence tasks.

| Method | 5 steps | 10 steps | 20 steps | 40 steps |
|---|---|---|---|---|
| L2P | 73.57 | 73.10 | 70.35 | 66.02 |
| $w$ KAC | 73.56 (-0.01) | 73.14 (+0.04) | 72.11 (+1.76) | 69.74 (+3.72) |
| DualPrompt | 74.57 | 72.48 | 70.68 | 66.31 |
| $w$ KAC | 76.37 (+1.80) | 75.67 (+3.19) | 74.68 (+4.00) | 71.24 (+4.93) |
| CODAPrompt | 77.62 | 77.01 | 74.40 | 69.34 |
| $w$ KAC | 80.14 (+2.52) | 79.24 (+2.23) | 77.94 (+3.54) | 74.31 (+4.97) |
| CPrompt | 78.68 | 76.80 | 74.32 | 70.07 |
| $w$ KAC | 79.08 (+0.40) | 78.07 (+1.27) | 75.73 (+1.41) | 72.05 (+1.98) |

2022b), we split the DomainNet (Peng et al., 2019) dataset into 6 domains, classifying a total of 345 categories across all tasks. All experiments are conducted in a non-exemplar setting, with no old samples saved for new training.

**Implementation Details.** To validate the effectiveness of KAC, we selected four prompt-based CIL approaches L2P (Wang et al., 2022d), DualPrompt (Wang et al., 2022c), CODAPrompt (Smith et al., 2023) and CPrompt (Gao et al., 2024b) as baselines, all of which have achieved superior performance across various CIL benchmarks. These approaches leverage learnable prompts to extract information from pre-trained backbones and classify the extracted embeddings using linear classifiers. We directly replace their classifiers with KAC with their original hyperparameters to train the model, allowing for a comparison of the differences between classifiers. We implement all compared approaches with their official code and their original selected hyperparameters.

## 4.2 Experimental Results

**Experiments on ImageNet-R.** Tab. 1 compares the accuracy of the last task between the baseline methods and those with KAC in the ImageNet-R settings. Replacing the linear classifiers with KAC leads to improvements across all methods, especially in challenging long-sequence scenarios, where gains of 3 to 5 points are observed in most cases. It demonstrates that KAC effectively helps models mitigate forgetting at each step. Furthermore, comparing CODAPrompt and CPrompt, we find that while both perform similarly when using linear classifiers, CODAPrompt outperforms CPrompt when switched to KAC. This indicates that the compatibility of KAC with different methods varies.

Table 2: Results on CUB200 dataset. The accuracy of the last task is reported. KAC delivers significant improvements for all baselines, especially in long-sequence tasks, highlighting its superior performance on fine-grained datasets.

| Method | 5 steps | 10 steps | 20 steps | 40 steps |
|---|---|---|---|---|
| L2P | 76.04 | 65.28 | 51.78 | 35.41 |
| $w$ KAC | 83.80 (+7.76) | 79.77 (+14.49) | 70.13 (+18.35) | 60.43 (+25.02) |
| DualPrompt | 76.38 | 64.60 | 54.68 | 37.55 |
| $w$ KAC | 85.03 (+8.65) | 79.61 (+14.01) | 71.91 (+17.23) | 64.69 (+27.14) |
| CODAPrompt | 78.73 | 71.87 | 58.00 | 37.81 |
| $w$ KAC | 85.61 (+6.88) | 82.59 (+10.72) | 73.32 (+15.32) | 64.56 (+26.75) |
| CPrompt | 82.02 | 76.80 | 72.99 | 64.80 |
| $w$ KAC | 83.08 (+1.06) | 80.75 (+3.95) | 78.54 (+5.55) | 76.51 (+11.71) |

**Experiments on CUB200.** Tab. 2 shows a comparison of the last accuracy of the last task in the CUB200 settings, surprising improvements achieving 10 to 25 percent are observed in long-

Table 3: Results on DomainNet. A Domain Incremental Learning experiment is conducted on it with 6 incremental domains of 345 classes. We report the average incremental accuracy and the accuracy of the last task. The results show that KAC can also work in DIL settings.

| Method | Linear | |
| --- | --- | --- |
| | Avg | Last |
| L2P | 57.78 | 49.22 |
| w KAC | 59.79 (+2.01) | 51.10 (+1.88) |
| DualPrompt | 60.96 | 51.83 |
| w KAC | 62.06 (+1.10) | 52.76 (+0.93) |
| CODAPrompt | 61.61 | 53.12 |
| w KAC | 62.78 (+1.17) | 53.54 (+0.42) |
| CPrompt | 61.32 | 52.49 |
| w KAC | 62.13 (+0.81) | 53.02 (+0.53) |

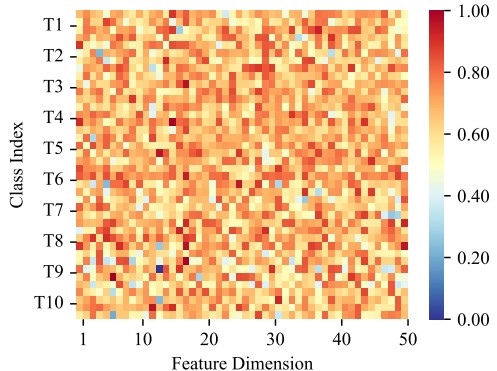

Figure 4: Activation maps for different classes across different channels. The x-axis represents 50 randomly selected channels from feature embeddings, while the y-axis represents classes from different tasks. The colors indicate varying levels of interest.

sequence scenarios. As CUB200 is a fine-grained bird classification dataset, we believe that KAC will perform well with such fine-grained datasets.

**Experiments on DomainNet.** We conduct experiments on DomainNet for Domain Incremental Learning, aiming to validate the ability of KAC to extend to other continual classification tasks. As shown in Tab. 3, when all approaches are implemented with KAC, the performance achieves an improvement of about 1 percent in average incremental accuracy and about 0.5 percent in last accuracy, demonstrating the robustness of our KAC.

**Visualization of activation maps.** Fig. 4 illustrates how different classes activate distinct channels, the differences in attention across different channels for various classes highlight the locality advantage in mitigating catastrophic forgetting, while all the activations remain stable during incremental tasks.

### 4.3 ABLATION STUDY

**Ablation on the number of basis functions.** The number of basis functions $N$ is a key hyperparameter of KAC. An excessive number of basis functions may lead to additional computations and result in a significantly high dimensionality of $W$. Conversely, a small $N$ may weaken the approximation ability of KAC. To explore an appropriate value for $N$, we conduct an ablation study on it. Fig. 5 shows the average incremental accuracy for four approaches using KAC with different numbers of basis functions in the 20 steps experiment on ImageNet-R. It indicates that simply increasing the number of basis functions does not benefit mitigating forgetting. Most approaches exhibit better performance when $N = 4$ or $N = 8$, encourages us to set $N$ as 4 in our experiments.

**The impact of structure over complexity.** To demonstrate that the advantages of KAC lie in the introduced KAN structure, not the additional computations, we replace the RBFs with an MLP layer, setting its output dimension to $N \times n$ to align the number of parameters with KAC using RBFs, allowing us to make a fair comparison between the two structure. Tab. 4 shows the performance of replacing RBFs with the MLP structure implemented on CODAPrompt. Upon comparison, we discover that whether the additional MLP structure is updated alongside the model or not, it does not yield any positive effects. This indicates that the advantages of KAC stem from its KAN structure rather than a simple increase in the dimensionality of the classification space.

## 5 CONCLUSIONS

In this paper, we explore the application of Kolmogorov-Arnold Networks (KAN) in continual learning and develop a novel continual classifier, the Kolmogorov-Arnold Classifier (KAC) which

Table 4: Ablation study on the structure of the classifier. We replace the spline functions in KAC with MLPs to validate the effectiveness of the KAN structure. Here, $w$ MLP represents the MLP trained alongside the model, while $w$ MLP (fixed) represents the randomly initialized MLP projection without any updating. The experiments are conducted in the 20 steps ImageNet-R scenario.

| | Avg | Last |
|---|---|---|
| CODAPrompt | 80.92 | 74.40 |
| $w$ KAC | 83.59 | 77.94 |
| $w$ MLP | 80.56 | 73.59 |
| $w$ MLP (fixed) | 65.87 | 51.03 |

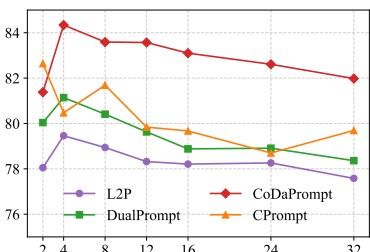

Figure 5: Ablation study on different numbers of basis functions in the 20 steps ImageNet-R scenario. The x-axis represents the number of basis functions, while the y-axis indicates the average incremental accuracy with varying numbers.

leverages KAN's inherent locality capability to reduce feature shifts during the learning process of new tasks. Analysis revealed that the poor approximation ability of the B-spline functions in KAN on high-dimensional data forced the model backbone to generate more shifts to approximate new classes. To address this issue, we introduce RBFs to replace the spline functions in KAN. KAC demonstrates significant advantages across various continual learning approaches and scenarios, showcasing its effectiveness and robustness. In the future, we will explore more possibilities of KAN in continual learning, fully leveraging its inherent advantages.

**Reproducibility Statement.** To ensure the reproducibility of our work, all data follows a standardized preprocessing pipeline, similar to the methods employed in CODAPrompt (Smith et al., 2023). We will also release all code to facilitate easy reproduction of our approach.

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
