## A   MORE IMPLEMENTATION DETAILS

All experiments are conducted with ViT-B/16 backbones. The numbed of RBFs is set as $N = 4$ and the centers $[c_1, c_2, \cdots, c_N]$ are evenly distributed between -2 and 2, and the $\sigma$ of the Gaussian functions is set as 1.3, allowing for an average division of the range from -2 to 2.

## B   MORE EXPERIMENTAL RESULTS

**Experiments on CIFAR-100.**   Tab. 5 compares the average incremental accuracy between the baseline methods and those with KAC in the CIFAR-100 dataset. Replacing the linear classifiers with KAC improves most of the methods, with a little drop in CPrompt and L2P. Due to the low pixel resolution of CIFAR-100, it is generally suitable for training smaller-scale networks. For pre-trained backbones, performance tends to be saturated, which is why our method does not show significant improvement on this dataset.

Table 5: The average incremental accuracy of CIFAR-100 10 steps scenario.

| Method | Linear | KAN CLF |
|---|---|---|
| L2P | 83.78 | 83.71  (-0.07) |
| DualPrompt | 84.80 | 85.74  (+0.94) |
| CODAPrompt | 86.65 | 87.26  (+0.61) |
| CPrompt | 87.50 | 87.19  (-0.31) |

**More results on CUB200 and ImageNet-R.**   Tab. 7 and Tab. 6 report the average incremental accuracy and the accuracy of the last task, in which most results demonstrate that with KAC, the approaches will achieve an improvement prepared with linear classifier.

Table 6: The average incremental accuracy and the accuracy of the last task in scenarios on ImageNet-R dataset.

| Model | 5 steps | | 10 steps | | 20 steps | | 40 steps | |
|---|---|---|---|---|---|---|---|---|
| | Avg | Last | Avg | Last | Avg | Last | Avg | Last |
| L2P | 78.42 | 73.57 | 79.58 | 73.10 | 77.93 | 70.35 | 74.28 | 66.02 |
| +KAN | 77.98 | 73.56 | 79.22 | 73.14 | 78.94 | 72.11 | 76.34 | 69.74 |
| DualPrompt | 79.75 | 74.57 | 79.50 | 72.48 | 78.35 | 70.68 | 74.51 | 66.31 |
| +KAN | 79.96 | 76.37 | 80.72 | 75.67 | 80.40 | 74.68 | 76.87 | 71.24 |
| CODAPrompt | 82.27 | 77.62 | 82.49 | 77.01 | 80.92 | 74.40 | 76.80 | 69.34 |
| +KAN | 83.75 | 80.14 | 84.43 | 79.24 | 83.59 | 77.94 | 79.79 | 74.31 |
| CPrompt | 84.07 | 78.68 | 83.13 | 76.80 | 81.83 | 74.32 | 78.98 | 70.07 |
| +KAN | 84.51 | 79.08 | 83.97 | 78.07 | 82.56 | 75.73 | 80.89 | 72.05 |

## C   MORE ABLATION STUDIES

**Ablation on the linear shortcut.**   In KAC, we don't follow conventional KAN, in which a linear shortcut is added with the spline functions. In this section, we show that the linear shortcut cannot help KAC achieve better performance. Tab. 8 reports the accuracy of the last task in ImageNet-R 20 steps scenario. It demonstrates that when linear shortcut is added, it achieves even worse accuracy, supporting our decision to remove the linear shortcut.

Table 7: The average incremental accuracy and the accuracy of the last task in scenarios on CUB200 dataset.

| Model | 5 steps | | 10 steps | | 20 steps | | 40 steps | |
|---|---|---|---|---|---|---|---|---|
| | Avg | Last | Avg | Last | Avg | Last | Avg | Last |
| L2P | 80.05 | 76.04 | 74.02 | 65.28 | 63.31 | 51.78 | 46.84 | 35.41 |
| +KAN | 84.42 | 83.80 | 81.54 | 79.77 | 73.70 | 70.13 | 66.08 | 60.43 |
| DualPrompt | 81.84 | 76.38 | 75.10 | 64.60 | 66.89 | 54.68 | 50.61 | 37.55 |
| +KAN | 86.20 | 85.03 | 82.18 | 79.61 | 76.93 | 71.91 | 71.31 | 64.69 |
| CODAPrompt | 83.09 | 78.73 | 79.30 | 71.87 | 69.49 | 58.00 | 52.57 | 37.81 |
| +KAN | 86.56 | 85.61 | 85.04 | 82.59 | 77.23 | 73.32 | 71.36 | 64.56 |
| CPrompt | 88.62 | 82.02 | 85.77 | 76.80 | 83.97 | 72.99 | 77.34 | 64.80 |
| +KAN | 89.60 | 83.08 | 89.04 | 80.75 | 87.06 | 78.54 | 85.11 | 76.51 |

Table 8: Comparison of the KAC and KAC with shortcut on the last accuracy in ImageNet-R 20 steps scenario.

| Method | Baseline | KAC + Shortcut | KAC |
|---|---|---|---|
| L2P | 70.35 | 71.09 | 72.11 |
| DualPrompt | 70.68 | 72.83 | 74.68 |
| CODAPrompt | 74.40 | 75.57 | 77.94 |
| CPrompt | 74.32 | 73.55 | 75.73 |

## D MORE VISUALIZATIONS

**Visualization of performance on CUB200.** To investigate the reasons behind the superior performance of KAC on CUB200, we make an observation on the accuracy curves of CUB200 across experiments with different steps. As shown in Fig. 6, with the arriving of tasks, KAC demonstrates a growing advantage. In several steps, the baseline frequently experiences significant forgetting, while KAC often exhibits less forgetting compared to the linear classifier during these steps, which helps KAC accumulate a higher final accuracy.

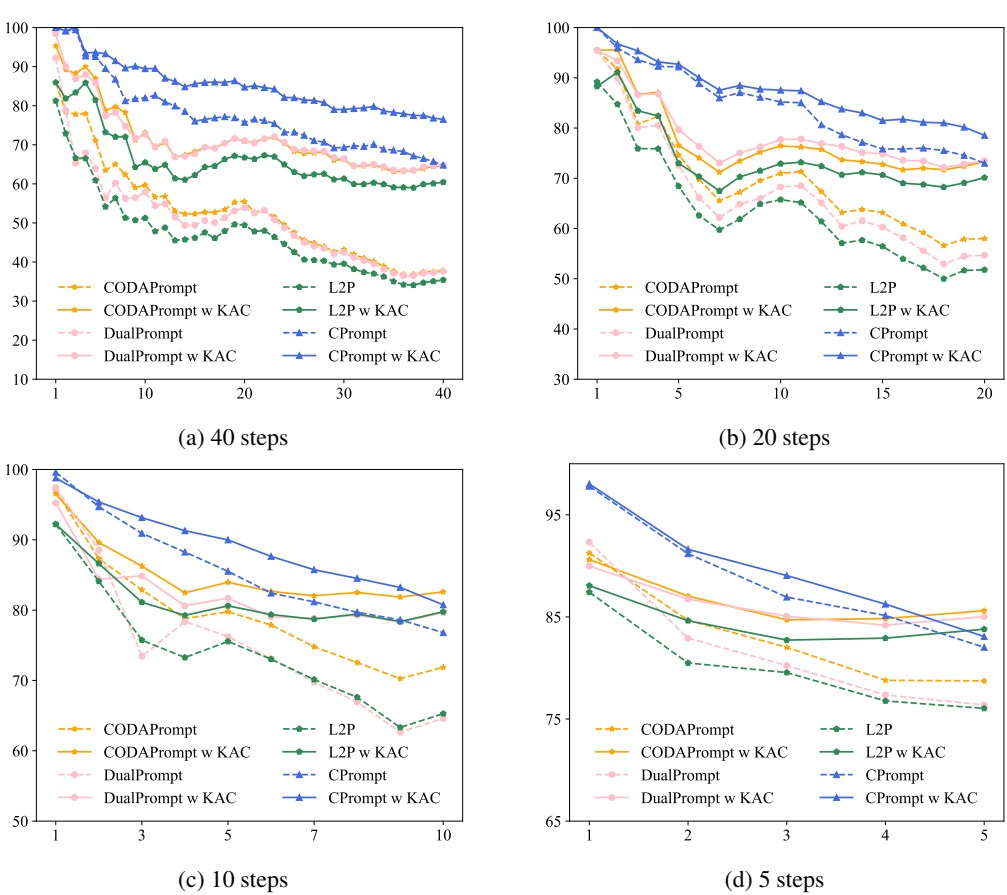

Figure 6: The accuracy curves for scenarios of different steps on the CUB200 dataset. The x-axis represents the gradually increasing tasks and the y-axis represents accuracy at each step. It can be observed that KAC follows the same trend as the baseline, but exhibits less forgetting at each step.