# OpenReview forum: "KAC: Kolmogorov-Arnold Classifier for Continual Learning"
_ICLR.cc/2025/Conference — ICLR 2025 Conference Withdrawn Submission_

### Official Review · Reviewer_8fEf · 2024-10-25

**Soundness:** 3
**Presentation:** 3
**Contribution:** 2
**Rating:** 5
**Confidence:** 4

**Summary:**

This paper proposes to improve existing state-of-the-art popular prompt-learning continual learning techniques. To do so, rather than improving the prompt design, the authors focus on the last fully connected layer commonly used for classification with such techniques. Therefore, this paper leverages Kolmogorov-Arnold Networks which learn class-specific activation functions. Precisely, they introduce a Kolmogorov-Arnold Classifier which relies on Radial Basis Functions rather than splines. Eventually, the paper shows significant improvement in the Class-Incremental Learning problem, especially over long task sequences.

**Strengths:**

- the paper is well written and overall easy to follow
- the problem of improving last layer learning of prompt-based methods is valuable to the CL community
- the usage of KAN is relatively intuitive and makes sense in this context
- the adaptation of KAN to more complex CL problems seems to be in its infancy
- the presented results show impressive improvements
- the ablation study is appreciated, especially the comparison to a regular MLP; though it could be improved

**Weaknesses:**

### Major weaknesses
- Various claims in this paper lack substantial proof. For instance:
	- l.502 "leverages KAN’s inherent locality capability to reduce feature shifts during the learning process of new tasks." or l.12 "11. Most existing methods utilize linear classifiers, which struggle to maintain a stable classification space while learning new tasks". While this might be true, I do not see any evidence in the paper for both claims.
	- l.82 "We find that the conventional KAN with B-spline functions struggles with high-dimensional data, leading to inadequate model plasticity, which may weaken the models’ plasticity when directly introduced as a classifier." Same remark here, the drop in average accuracy displayed in Figure 2 can also be the consequence of a lack of stability. Plasticity-focused metrics should be used. This also applies to lines 223 and 227.
	- Figure 1 caption: "Conventional linear classifiers activate each weight equally across all tasks, whereas our Kolmogorov-Arnold Classifier learns class-specific learnable activations for each channel across all categories, minimizing forgetting caused by irrelevant weight changes". There is no proof of such behavior here.
- The experiments are quite limited in terms of reported metrics and number of runs. In CL especially, the seed used can have a significant impact, thus I strongly encourage the authors to run several experiments and report the average and standard deviations.
- Reporting only the last task accuracy on Table 1 and Table 2 is very biased toward plasticity. However, as mentioned, this paper claims a huge gain in stability. Similarly, the last task's accuracy is very dependent on the seed used.
- The analysis of why the methods perform especially well on long sequences is lacking. The authors claim l. 411 "It demonstrates that KAC effectively helps model mitigate forgetting at each step". However, as pointed out above, it is only the final task accuracy, so this narrative is not proven.

### Minor weaknesses
- In equations 5, 6, 7, 8, and 10, if my understanding is correct, $x_p$ is a scalar and $c_i$ a vector. I think you should clarify this operation.
- In Figure 3, you first use a stack of 1d Gaussians and then 3d Gaussians after weighting. Since it represents almost the same thing (just with different weights), in the same diagram, I personally find it a bit confusing.
- l.229 "This is because splines cannot effectively model the compositional structure present in the data, while linear classifiers benefit from their fully connected structure, allowing them to learn this structure effectively (He & Xu, 2023)." If my understanding is correct, the KAN architecture is also fully connected (every output depends on every input). So how is that a beneficial property specific to the linear classifier?
- Figure 4 is unclear. Is this the activation of randomly selected inputs of each class? The description l.460 could be improved. A visualization of the learned activation functions for some features across various tasks could be interesting.

### typos
- $x_p$ font not consistent in equation 6. Same in equation 8, and 9 and 11 (for $x$).
- figure 2 and figure 5, legends would be better on the figure rather than caption.

**Questions:**

- How are $c_i$ initialised? What is the impact of the initialisation on the performances and/or convergence?
- How do you explain the results of Figure 5?
- I appreciate the comparison with an MLP. However, I think it lacks details.
	- Did you use logits masking? This is used for most CL methods (such as CODA) and greatly increases performances.
	- "Tab. 4 shows the performance of replacing RBFs with the MLP structure implemented on CODAPrompt" By default they use a linear layer. Do you mena they also have an MLP implementation?
	- Do you also use the layer norm? What is the intuition behind the usage of layer norm? Did you make an ablation study of this?

---

### Official Review · Reviewer_6DpJ · 2024-10-29

**Soundness:** 2
**Presentation:** 3
**Contribution:** 2
**Rating:** 3
**Confidence:** 4

**Summary:**

In this paper, the authors introduce the Kolmogorov-Arnold Classifier (KAC), a novel classifier developed for continual learning based on the KAN structure.

**Strengths:**

This paper employs the RBF-based KAN to present a new classifier for continual learning methods. The experiments show the advantages of the proposed methods over the benchmark. The writing is clear, and the narrative is easy to follow, facilitating understanding complex concepts.

**Weaknesses:**

The originality of this work is low. The KAN has been used for continual learning in ‘KAN: Kolmogorov–Arnold Networks’. Also, the RBF has been employed for the KAN in ‘Kolmogorov-Arnold Networks are Radial Basis Function Networks’.

The experiments are insufficient. For example, no Monte Carlo experiments measure the mean and deviation of the prediction accuracy. Also, there is no average prediction accuracy over all tasks in the CL experiments.

**Questions:**

1)	Please clarify the calculation of equation (7) by adding equation (2.5) in ‘KAN: Kolmogorov–Arnold Networks’. Also, explain the meaning of n_l, n_l+1.
2)	Lines 232-234 explain the stacking KAN layers can mitigate COD. Why not directly employ a multi-layer KAN for the classifier design? Further, please explain compared to a pure multi-layer KAN, what is the advantage of the proposed framework for CL?
3)	The reviewer is sceptical about the correctness of equation (7). The linear combination of Gaussian densities is not a Gaussian density.
4)	The reviewer disagrees with the sentence on Line 305, where it says f(x) conforms to a Gaussian process. However, f(x) is a deterministic function but not a stochastic model.
5)	More experiment results are needed. In all experiment results, no Monte Carlo experiments measure the mean and deviation of the prediction accuracy. Also, there is no average prediction accuracy over all tasks. This measure is important to judge the performance of a CL method in achieving the balance between stability and plasticity. Instead, only average incremental accuracy and last task accuracy are provided.
6)	On line 505-507, it says ‘To address this issue, we introduce RBFs to replace the spline functions in KAN. KAC demonstrates significant advantages …’. However, the RBF has been employed for the KAN in ‘Kolmogorov-Arnold Networks are Radial Basis Function Networks’. The description needs to be refined.

---

### Official Review · Reviewer_X6TA · 2024-11-02

**Soundness:** 3
**Presentation:** 3
**Contribution:** 3
**Rating:** 5
**Confidence:** 3

**Summary:**

This work introduces a novel classifier based on the Kolmogorov-Arnold network (KAN) structure, which aims to solve the catastrophic forgetting problem in continual learning. The authors propose to replace the traditional linear classifier with KAC using radial basis functions (RBF) to enhance the plasticity and stability of the model. Experiments demonstrate the effectiveness of KAC in improving performance and reducing forgetting. The main contributions include the exploration of KAN in continual learning, the introduction of KAC using RBF, and the verification of its performance on multiple benchmarks.

**Strengths:**

- **Originality**: This paper proposes a novel continual learning method by adapting the Kolmogorov-Arnold network, which is a unique insight to solve the catastrophic forgetting problem. Replacing the B-spline function with RBF is also a creative solution that can improve the performance of the model in high-dimensional data scenarios.
- **Clarity**: The paper is well organized and easy to follow. The figures and tables are informative and support the text well.
- **Quality**: The experiments are well designed and conducted on multiple datasets to thoroughly evaluate the proposed KAC. However, there is still room for improvement

**Weaknesses:**

- **Time Complexity and Computational Cost**: The paper does not provide a detailed analysis of the time complexity and computational cost associated with the Gaussian kernel-based KAN method compared to traditional MLPs, ***like linear classifiers***. This is a significant weakness as the practicality of the approach could be limited by increased resource requirements.
- **Lack of Comparison with Traditional Continual Learning Methods**: The paper primarily compares the KAC with prompt-based approaches. There is a lack of comparison with more traditional continual learning methods, ***such as EWC [1]***, which could provide a more comprehensive understanding of the KAC's performance relative to the state-of-the-art.

[1] Kirkpatrick, James, et al. "Overcoming catastrophic forgetting in neural networks." Proceedings of the national academy of sciences 114.13 (2017): 3521-3526.

（2024/11/14: The parts marked in bold and italics are some newly added specific examples for comparison, which can be used as a reference for authors.）

**Questions:**

1. How does the KAC perform in other domains or with different types of data? Are there any scalability issues that could limit its application in larger or more diverse datasets?
2. Can the authors discuss potential limitations or considerations regarding the deployment of the KAC in real-world scenarios, especially concerning resource utilization?
3. Are there plans to validate the robustness of the KAC against adversarial attacks or in other stress tests that could challenge the stability and plasticity of the model?

---

### Note · Authors · 2024-11-14

I have read and agree with the venue's withdrawal policy on behalf of myself and my co-authors.